# Normative Structure of Resting-State EEG in Bipolar Derivations for Daily Clinical Practice: A Pilot Study

**DOI:** 10.3390/brainsci13020167

**Published:** 2023-01-18

**Authors:** Jesús Pastor, Lorena Vega-Zelaya

**Affiliations:** Clinical Neurophysiology and Instituto de Investigación Biomédica, Hospital Universitario de La Princesa, C/Diego de León 62, 28006 Madrid, Spain

**Keywords:** coherence, double-banana bipolar montage, fast Fourier transform, normative equations, posterior dominant rhythm, qEEG, Shannon’s spectral entropy

## Abstract

We used numerical methods to define the normative structure of resting-state EEG (rsEEG) in a pilot study of 37 healthy subjects (10–74 years old), using a double-banana bipolar montage. Artifact-free 120–200 s epoch lengths were visually identified and divided into 1 s windows with a 10% overlap. Differential channels were grouped by frontal, parieto-occipital, and temporal lobes. For every channel, the power spectrum was calculated and used to compute the area for delta (0–4 Hz), theta (4–8 Hz), alpha (8–13 Hz), and beta (13–30 Hz) bands and was log-transformed. Furthermore, Shannon’s spectral entropy (SSE) and coherence by bands were computed. Finally, we also calculated the main frequency and amplitude of the posterior dominant rhythm. According to the age-dependent distribution of the bands, we divided the patients in the following three groups: younger than 20; between 21 and 50; and older than 51 years old. The distribution of bands and coherence was different for the three groups depending on the brain lobes. We described the normative equations for the three age groups and for every brain lobe. We showed the feasibility of a normative structure of rsEEG picked up with a double-banana montage.

## 1. Introduction

The electroencephalogram (EEG) is the multivariate spatiotemporal determination of the electrical potentials generated by the brain and recorded on the surface of the scalp. EEG is the electrical recording obtained at the scalp surface from a linear superposition of all the neural sources inside the brain [1]. The online recorded signal by one EEG channel is the potential difference between two electrodes, leaving the choice of reference electrode undetermined [2]. Ideally, a reference electrode must be placed where zero potential can be recorded, which would imply its placement at an infinite distance [1,2]. Obviously, this is a factual impossibility, and a physical reference must be located at or near the scalp. Nevertheless, all of the electrodes placed at the scalp surface would be affected by the linear superposition; therefore, the picked-up signals would be correlated to the signals recorded by the active electrodes [3].

Several attempts for a so-called quiet reference have long engaged many EEG scientists and clinicians. Their purpose was to record scalp potentials that are essentially monopolar in nature, imagining the presence of bioelectrical sources directly under the so-called monopolar electrode [1,2]. In theory, if the putative reference location can be eligible as a genuine reference, several alternative reference sites should be easily available, and the EEG should not change substantially as a result of reference changes to alternative sites [2].

Previous notable attempts to identify the optimal reference have been carried out. Among these, a cephalic electrode in the midline (e.g., FCz, Fz, Cz or Oz), linked mastoid electrodes [4] (LM), and the average reference [5,6] (AR), as well as the reference electrode standardization technique [7] (REST) and its regularized version [8] (rREST), have been used.

A different approach to the problem of reference is to dispense this question using non-unipolar references as bipolar or scalp Laplacian (SL) montages. Although bipolar reference recordings are not the way to obtain the true scalp potential but rather to show the local surface potential variance of underlying brain sources as the first derivative of potentials [2], it is the montage most often used in clinical practice [9,10,11,12,13,14,15,16,17,18]. However, other clinical fields of research, such as neurofeedback or event-related potentials (ERPs), generally use unipolar recordings [19,20].

Although several articles have described the normative values of unipolar recordings from infants to elderly individuals [21,22,23], less work has been performed on bipolar derivations. In fact, only eight derivations of the classical double-banana montage were described in the seminal work of John et al. [24], despite the thorough use of this montage. However, the values for spectra or synchronization are different for unipolar and non-unipolar montages [2,3]. Considering that the last montage is the most used in routine clinics, we should have normative values obtained for the bipolar montage to use in daily clinical practice but, until now, these values have not been described.

The structure can be defined as the arrangement of and relationship between the parts or elements of something complex. We can apply this concept to the EEG as the arrangement and relationships of the different bands (delta, theta, alpha, and beta) and synchronization across the whole scalp. In this pilot study, we analyzed the feasibility of building a normative structure of resting-state EEG (rsEEG) recorded in a double-banana bipolar montage (the most frequently used in clinical practice) in healthy people using the common conditions used in daily clinical practice, e.g., eye-closed recordings performed in the same environment as an EEG performed at a tertiary hospital devoted primarily to clinical assistance, using standard 19 electrodes placed according to the 10–20 International System electrode cap. Our aim is to describe the equations fitting the topographical values across the scalp or spectra and synchronization as a function of age. The description of normative values is the first essential step in identifying the numerical features of pathological conditions.

Abbreviations used can be found at the end of the work.

## 2. Materials and Methods

### 2.1. Subjects

This study evaluated 10 (29.1 ± 4.8) men and 27 women (32.3 ± 3.2 years old) from 2018 to 2022. The experimental procedure was reviewed by the medical ethical review board of the Hospital Universitario de La Princesa; however, considering the harmlessness of the EEG in well-informed voluntary subjects, no specific code was necessary. The purpose of this study was explained to all the subjects and written or oral informed consent was obtained from all of them. In the case of subjects under 18 years old, a family member approved their participation in the study.

The subjects ranged from 10 to 73 years old and none of them suffered from known mental or neurological illnesses. None of them took drugs affecting the central nervous system (CNS), although some of them were treated with anti-hypertensive, non-steroid anti-inflammatory, or gastric-protecting drugs. All of them were right-hand dominant.

### 2.2. EEG Recording and Numerical Analysis

Eye-closed rsEEG records were obtained while the subjects were seated comfortably in a sound- and light-attenuated room, using a 32-channel digital system (EEG32U, NeuroWorks, XLTEK^®^, Oakville, ON, Canada) with 19 electrodes placed according to the 10–20 International System. In addition, the differential derivation I of Einthoven for the ECG was positioned. Recordings were performed at a 512 Hz sampling rate, with a filter bandwidth of 0.5 to 70 Hz and a notch filter of 50 Hz. Electrode impedances were below 20 kΩ. A raw record of at least 180 s free of artifact was selected for analysis.

We had two branches of analysis: power spectra and synchronization. For both of them, we had dynamical (i.e., varying with time) and mean measurements (i.e., mean spectra or mean graph of synchronization).

The process used for qEEG follows the next steps:

Different length raw records (minimum of 5 min) were exported from the EEG device to an ASCII file. Artifacts were excluded by the export of several artifact-free chunks, which were later put together for analysis. Although the raw recordings were digitized at 512, we down sampled to 128. Exported files were digitally filtered by a sixth-order Butterworth digital filter between 0.5 and 30 Hz.

The differential EEG double-banana montage was reconstructed. Topographic placement of channels was defined on the scalp as the midpoint between the electrode pairs defining the channel, e.g., the Fp1–F3 channel was placed at the midpoint of the geodesic between the Fp1 and F3 electrodes.

All recordings were divided into 1 s moving windows with a 10% overlap. The total length used during the fast Fourier transform (FFT) was directly related to frequency precision in the power spectrum (PS). Overlap was used to minimize the effect produced by windowing [25]. 

For each window (*n*) and frequency (*k*), we computed the discrete FFT of the voltage ($V^{m}\left(n\right))$ obtained from every channel (*m*) to obtain the power spectrum ($S_{n,k}^{m}$, in µV^2^/Hz). We used this expression:(1)$$S_{n,k}^{m}=\sum_{n=0}^{N-1}V^{m}\left(n\right)e^{-i\frac{2\pi }{N}kn};m=Fp_{1},\;F_{3},\ldots$$

We also computed Shannon’s spectral entropy (SSE) according to
(2)$$SSE_{k}^{m}=-\sum_{k=0}^{F}p_{k}\log_{2}p_{k}$$
where *F* is the maximum frequency computed and *p_k_* is the probability density of *S*, obtained from the expression
(3)$$p_{k}=\frac{S_{n,k}^{m}}{\sum_{k=0}^{F}S_{n,k}^{m}\Delta k}$$

We computed the area under $S_{n,k}^{m}$ according to the classical segmentation of EEG bands (in Hz) as follows: delta (*δ*), 0.5–4; theta (*θ*), 4–8; alpha (*α*), 8–13; and beta (*β*), 13–30. We used the following expression:(4)$$A_{j}\left(k\right)=\sum_{k=inf}^{sup}\;S_{n}^{m}\left(k\right)\Delta k;j=\delta ,\;\theta ,\alpha ,\beta$$

The expression sup refers to the upper limit of every EEG band and where Δ*k* is the increment of frequency.

The different EEG bands were rooted in different neural systems; therefore, the synchronization of different bands can offer specific information. A very useful method used to assess specific bands is coherence [26,27], which is defined by the following equation:(5)$$Coh\left(\omega \right)=\frac{{\left|\langle S_{ij}\left(\omega \right)\rangle_{n}\right|}^{2}}{\langle S_{ii}\left(\omega \right)\rangle_{n}\langle S_{jj}\left(\omega \right)\rangle_{n}}$$
where *S_ij_* is the cross-spectrum of channels *i* and *j*, normalized by the power spectra *S_ii_* and *S_jj_*. The symbol 〈. . .〉_n_ means the average over *n* epochs. The averaging of the cross-spectrum *S_ij_* must be performed before the absolute value is taken [25]. We can observe that the spectra and cross-spectrum are functions of the frequency.

The mean value of all windows was computed, obtaining the mean correlation and coherence matrices.

Areas of the same band were grouped by cerebral hemisphere and lobe. In the case of the left hemisphere (shown as an example), we grouped for frontal $F=\left\{\frac{\left(Fp_{1}-F_{3}\right)+\left(F_{3}-C_{3}\right)+\left(Fp_{1}-F_{7}\right)}{3}\right\}$, parieto-occipital $PO=\left\{\frac{\left(C_{3}-P_{3}\right)+\left(P_{3}-O_{1}\right)+\left(T_{5}-O_{1}\right)}{3}\right\}$, and temporal $T=\left\{\frac{\left(F_{7}-T_{3}\right)+\left(T_{3}-T_{5}\right)+\left(T_{5}-O_{1}\right)}{3}\right\}$. For the whole left hemisphere, we used the expression $H=\left\{\frac{\left(Fp_{1}-F_{3}\right)+\left(F_{3}-C_{3}\right)+\left(Fp_{1}-F_{7}\right)+\left(C_{3}-P_{3}\right)+\left(P_{3}-O_{1}\right)+\left(T_{5}-O_{1}\right)+\left(F_{7}-T_{3}\right)+\left(T_{3}-T_{5}\right)+\left(T_{5}-O_{1}\right)}{9}\right\}$. Channels from the right hemisphere were grouped accordingly. These areas, for both bands (*j*) and lobes (*r*), $A_{j}^{r}\left(t\right);\;r=H,F\;PO,\;T$, were plotted as time functions and compared between hemispheres. The same groups were used to compute *SSE.*

The mean values of the synchronization for the hemispheres and lobes were computed as the average of all pairs ($N_{pairs}$) of channels ($N_{ch}$), according to this expression $N_{pairs}=\frac{N_{ch}\left(N_{ch}-1\right)}{2};N_{ch}=3\;for\;lobes\;and\;N_{ch}=9$ for hemispheres.

The numerical analysis of EEG recordings was performed with custom-made MATLAB ^®^ R2018 software (MathWorks, Natick, MA, USA).

### 2.3. Statistics

Evidently, the absolute values of PS are quite different from patient to patient; therefore, the comparison with raw values is not useful. Therefore, we used log transformation [28] for the PS measures (logPS). The synchronization measures and *SSE* were compared by the raw values.

Posterior dominant rhythm (PDR, in Hz) was measured at the peak of the alpha band in channels P3-O1 and P4-O2 for the left and right hemispheres [29]. With the windowing used, the precision in frequency measurement was 0.5 Hz. In the same channels, we measured the amplitude (*V*, in *µV*) by means of the root mean square, according to this expression
(6)$$V_{j}=\sqrt{\frac{\sum_{i=1}^{N}x_{i}^{2}}{N}};j=left,\;right$$
where *x_i_* represents the discrete values of voltage for a numerical series of *N* points.

The descriptive statistic is offered as mean ± SEM.

Statistical comparisons between group values picked from symmetric regions (e.g., left and frontal lobes) were performed by the paired Student’s *t*-test or Wilcoxon signed-rank test. Normality was evaluated using the Kolmogorov–Smirnov test. 

Quadratic nonlinear regression was performed by means of the least-squared method, and the correlation coefficient (ρ) was used to study the dependence between variables. Statistical significance was evaluated by means of a contrast hypothesis against the null hypothesis ρ = 0 using the formula
(7)$$t=\frac{r\sqrt{n-2}}{\sqrt{1-r^{2}}}$$

This describes a one-tailed Student’s *t*-test distribution with *n* − 2 degrees of freedom [30].

Test–retest reliability was assessed by means of Pearson’s correlation coefficient in a subset of 17 patients (aged between 10 and 45 years old) for periods between one week and two weeks after the first measurement. 

SigmaStat^®^ 3.5 software (SigmaStat, Point Richmond, CA, USA) and MATLAB^®^ R2018 (MathWorks, Natik, MA, USA) were employed for statistical analysis. 

The significance level was set at *p* = 0.05. 

## 3. Results

We assessed the test–retest reliability for all the measures of logPS, *SSE* and coherence, and PDR. Coefficients of stability for test–retest reliability were higher for the logPS measurements (0.9846 ± 0.0038), followed by PDR (0.9690 ± 0.004 for frequency and 0.9202 ± 0.0450 V), then followed by SSE (0.9010 ± 0.0045), and lower for coherence (0.8744 ± 0.0194). Therefore, the values obtained from the method were highly reliable.

### 3.1. EEG Power Spectrum Structure as Function of Age

We compared the logPS of all the bands between symmetric lobes using the paired Student’s *t*-test or the Wilcoxon signed-rank test when normality failed and we did not find any differences; therefore, the distribution of bands was symmetric across the scalp (Table 1), although the relationship between bands (i.e., δ, θ, α, and β) changed from lobe to lobe. 

We plotted the logPS as function of age for all the bands of brain lobes (Figure 1). The results were fitted to quadratic functions by means of the least-squared method. We observed that, except for the beta band at the left and right frontal lobes, the EEG bands (22/24) fitted very well to the quadratic functions. 

Therefore, we can use these functions as equations (see Appendix A) to obtain the normative values. The functions are shown in Appendix A.

The structure of bands not only depends on age but is also lobe-dependent, although symmetric between the left and right hemispheres. To systematize the structure of bands for every lobe, we divided the whole range of age into three periods, i.e., younger than 20 (denoted as <20, *N* = 7), between 20 and 50 (denoted as 20–50, *N* = 25), and older than 50 years old (denoted as >50, *N* = 5). These ranges were chosen because the relationship between bands were stable during all periods, i.e., no crossing of regression functions was observed for all periods. The structure for the different lobes changed as a function of the age as can be seen from the Table 2. For the frontal lobe, the structures were $F_{<20}=\left\{\delta ,\theta ,\alpha ,\beta \right\}$, $F_{20-50}=\left\{\delta ,\beta ,\alpha ,\theta \right\}$, and $F_{>50}=\left\{\delta ,\alpha ,\beta ,\theta \right\}$, respectively, for the <20, 20–50, and >50 years groups. For parieto-occipital lobes, the structures were $PO_{<20}=\left\{\alpha ,\delta ,\theta ,\beta \right\}$, $PO_{20-50}=\left\{\alpha ,\delta ,\beta ,\theta \right\}$, and $PO_{>50}=\left\{\alpha ,\beta ,\delta ,\theta \right\}$. Finally for the temporal lobes, the structures were $T_{<20}=\left\{\alpha ,\delta ,\beta ,\theta \right\}$, $T_{20-50}=\left\{\delta ,\alpha ,\beta ,\theta \right\}$, and $T_{>50}=T_{<20}$.

Therefore, for the F lobes, the dominant rhythms were δ, followed by θ in the younger group, which was substituted by faster ones (α and β) as age increased. Meanwhile, in the PO lobes, the most important component was α rhythm for all age groups. In the T lobes, α rhythm was prevalent for the younger and older groups, but δ was the dominant rhythm in the middle age group.

Usually, the lower values were found between 30–40 years for all the bands, except for the delta band which showed a lower value at years above 60. On the contrary, for δ, θ and α were always the highest values obtained for ages <20 years.

Another relevant measurement of the EEG used in clinical practice is the amplitude and the PDR, usually measured in the parieto-occipital channels (P3-O1 and P4-O2). We addressed the dependence of these variables on age. In Figure 2A, we shows that the PDR can be fitted to this function $\mathrm{PDR}\left(Hz\right)=9.77+0.03\times years-0.0004\times {\left(years\right)}^{2}-0.000002\times {\left(years\right)}^{3}$ (r = 0.2977, *p* < 0.05, Student’s *t*-test), with a maximum of 10.3 Hz at 35 years old. Fitting the data to a linear function, we obtained a lower value correlation (r = 0.1965; n.s).

We addressed the amplitude of the parieto-occipital channels as an age function (Figure 2B). No differences in amplitude were observed between both hemispheres and the data were very well fitted to negative exponential functions, namely, $V_{left}\left(\mu V\right)=14.62+99.31e^{0.127\times \left(years\right)}$ and $V_{right}\left(\mu V\right)=14.62+99.32e^{0.127\times \left(years\right)}$.

A way to characterize the complexity of PS is SSE. We did not observe differences between the right and left F lobes (*p* = 0.122, paired Student’s *t*-test), PO lobes (*p* = 0.084, paired signed-rank test), and T lobes (*p* = 0.098, paired Student’s *t*-test). Consequently, we addressed the relationship between entropy and age (Figure 3).

The quadratic regression functions were (probability for Student’s *t*-test) ${\mathrm{SSE}}_{left\;F}=3.2247+0.1128\times years-0.0027\times {\left(years\right)}^{2}+0.00002\times {\left(years\right)}^{3}$, r = 0.3242 (*p* < 0.01), ${\mathrm{SSE}}_{right\;F}=3.0516+0.1374\times years-0.0034\times {\left(years\right)}^{2}+0.00002\times {\left(years\right)}^{3}$, r = 0.315 (*p* < 0.01), ${\mathrm{SSE}}_{left\;PO}=3.3885+0.0726\times years-0.0012\times {\left(years\right)}^{2}+0.000005\times {\left(years\right)}^{3}$, r = 0.3718 (*p* < 0.01); ${\mathrm{SSE}}_{right\;PO}=2.7445+0.1229\times years-0.0025\times {\left(years\right)}^{2}+0.000015\times {\left(years\right)}^{3}$, r = 0.4638 (*p* < 0.001); ${\mathrm{SSE}}_{left\;T}=3.9949+0.0589\times years-0.0014\times {\left(years\right)}^{2}+0.0000096\times {\left(years\right)}^{3}$, r = 0.2515 (*p* < 0.05); ${\mathrm{SSE}}_{right\;T}=3.4080+0.0994\times years-0.0023\times {\left(years\right)}^{2}+0.000015\times {\left(years\right)}^{3}$, r = 0.3658 (*p* < 0.01). All the functions significantly fit the data, meaning that a relationship between age and the complexity of PS was observed.

The lowest values (i.e., the simplest structure of PS) of SSE were found at 10 years, increasing around 30–40 years, decreasing at 50–60 years and, finally, for the frontal lobes, increasing again, meanwhile SSE of the PO lobes decreases.

### 3.2. EEG Synchronization as a Function of Age

The age-dependent synchronization variations and variations across the scalp can be addressed and are needed to characterize the physiological structure of rsEEG (Figure 4). The comparison of all the measurements between the symmetric lobes showed no differences, except for alpha coherence at the temporal lobe (Cohα_T_), lower at the left temporal lobe, (0.201 ± 0.020/0.250 ± 0.024 for the left/right temporal lobes respectively, *p* = 0.016, Wilcoxon signed-rank test), and the alpha coherence of the whole hemisphere (Cohα_H_) (0.176 ± 0.009/0.186 ± 0.019 for left/right lobes respectively *p* = 0.043, Wilcoxon signed-rank test).

From an overall amount of 40 quadratic regression functions, only 17 (43%) fitted data with statistical significance (see Table A1 at Appendix B). 

It is interesting to observe that 9/17 (53%) functions fit data from the PO lobes, followed by regression functions from the T lobes at 4/17 (24%), then followed by the frontal F lobes at 3/17 (18%), and finally, hemispheric (H) regressions at 2/17 (12%). Therefore, synchronization from PO lobes was fitted for all the measurements, except for right Cohδ and Cohθ. It is interesting to observe that all the measures for Cohα, such as from the left and right hemispheres, showed statistically significant correlations. The rest of the synchronization measurements did not show a significant dependence on age. Those functions with a statistical significance are listed in Appendix B.

Therefore, we did not find an age dependence of synchronization as well defined as that for PS, except for the Cohα, which fits very well to the quadratic functions of all three groups.

The structure of synchronization across the scalp for the different groups of age is shown in Table 3.

For all age groups, the structure of synchronization at the frontal lobes was the same and symmetric between both sides, i.e., $H_{<20}=\left\{Coh\alpha ,Coh\delta ,Coh\beta ,Coh\theta \right\}=H_{>50}$; $H_{20-50}=\left\{Coh\delta ,Coh\theta ,Coh\alpha ,Coh\beta \right\}$. In the case of the frontal lobes, we have $F_{<20}=\left\{Coh\delta ,Coh\theta ,Coh\alpha ,Coh\beta \right\}=F_{20-50}=F_{>50}$. In the case of the PO lobes, the highest values of coherence were Cohα and Cohθ for all three age groups with lower values for Cohδ and Cohβ, so $PO_{<20}=\left\{Coh\alpha ,Coh\theta ,Coh\delta /\beta ,Coh\beta /\delta \right\}=PO_{20-50}=PO_{>50}$. Finally, for the temporal lobes, Cohδ and Cohα were the highest values in equal parts, $T_{<20}=\left\{Coh\delta /\alpha ,Coh\alpha /\delta ,Coh\theta /\beta ,Coh\beta /\theta \right\}=T_{20-50}=T_{>50}$.

On the contrary of PS, the Coh structure at a defined lobe/hemisphere was the same for all the three age groups.

## 4. Discussion

### 4.1. Summary and Contribution

In this work, we obtained the normative structure of rsEEG from a group of left-handed healthy subjects in a broad age range. We obtained the structure of bands for a double-banana bipolar montage, the PDR features and the synchronization by lobes, and the hemispheres using coherence. These quantitative measures depict the physiological structure of the EEG of healthy humans for the most used montage in daily clinical practice.

We used the assumption that EEG is founded in a homeostatic system [19,31,32]. This aspect is of extraordinary relevance because different pathologies lead to specific changes in the different numerical variables obtained. The complex neuroanatomical homeostatic system is probably genetically determined and regulates basal levels of local synchronization, global interactions between different regions, spectral composition, and periodic signal space sampling [19,33,34,35].

Until now, the normative values for rsEEG have been mostly reported for unipolar montages [21,22,23,36,37] and only partially for bipolar montages [24], limiting its usefulness in routine clinical practice. The normative equations developed by John et al. [24] stated that specific logarithmic and ratio transforms must be applied to all EEG power, EEG coherence, EEG phase, and EEG amplitude asymmetries to best approximate normal distribution. Nearly all the variables analyzed in this paper were well fitted to Gaussian distribution.

However, the way we obtain the EEG is relevant for the quantitative values of PS and synchronization variables. In fact, unipolar or non-unipolar references can be used for scalp voltage estimation. For routine clinical practice, unipolar and bipolar montages are easily interchangeable; however, from a quantitative perspective, they give rise to very different values of PS and synchronization. Therefore, it is of great importance to discriminate between the kinds of measurements performed.

Every montage has pros and cons. The physical LM has three main problems that limit its results as an optimal reference. In fact, the linked-ears reference does not generally accomplish an approximation of potential at infinity and may distort, although to a small degree [38], the surface potential maps; however, this reference violates the normal boundary conditions of zero-current flux from the scalp [39]. Finally, a serious problem is that if the two contact impedances in the linked path are unequal, the effective reference is unbalanced to one side [1], and the physical LM reference may actually be a random reference rather than a quiet reference. In fact, it was shown that LM seriously biases EEG power [40] and coherence spectra [41], confounding the interpretation of results [7]. In 1950, the first clinical use of AR was reported [5], assuming that if EEG sources consist of a large number of randomly placed and randomly oriented dipoles, a rather constant zero average will be obtained over the surface of the scalp. Experience with the average monopolar reference electrode shows that this is usually approached in practice [6]. The AR is currently one of the most widely accepted references, and it is now implemented by the off-line re-referencing of digital signals. After the demonstration that the surface integral of a dipole should be zero [42], AR was advocated as one of the best reference options [1]. In recent decades, the reference electrode standardization technique (REST) was proposed to approximately reconstruct EEG potentials with an infinite reference [43] and has since been evaluated extensively with various simulations [8,44]. Recently, it has been demonstrated that both REST and AR are particular cases of a unified reference estimator under the Bayesian statistical framework [45]. However, the use of the REST as a gold standard for unipolar recordings is still debated [21].

Clinically and pathophysiologically oriented studies preferentially use different montages, i.e., bipolar and unipolar, respectively. Additionally, there is a growing consensus that comparing the rsEEGs of individuals with a normative database to assess clinically meaningful deviations can be used for diagnostic procedures and guide personalized treatment effectiveness [46]. However, normative values must be obtained from an adequate montage because, otherwise the values cannot be straightforwardly used.

### 4.2. Strengh and Limitations

The use of unipolar montages is highly recommended in works with pathophysiological or neuroscientific orientation, such as high-density recordings [47], connectivity studies [48,49], response to pharmacological treatment [50,51,52], or the diagnosis of neurological or psychiatric pathologies [51,52,53,54,55]. Additionally, normative datasets with unipolar montages have been described in detail [56,57,58].

However, most works in clinical practice use the bipolar montage, although they generally do not discuss what a better montage should be. Others have used unipolar references for epilepsy [59,60,61], the intensive care unit [59,60,61,62,63,64], for the diagnosis of several other pathologies [65,66,67,68,69,70], mainly using qEEG [71,72,73,74], and even during psychological studies, addressing the presence of frontal-lobe alpha activity, and in psychiatric populations [75,76].

Although the number of subjects in our pilot study was low, most of the variables fit a Gaussian distribution. In fact, one would expect, based upon the central limit theorem, that EEG variables would approximate a normal distribution as the sample size increases, assuming no artefact or experimenter bias [77].

The structure of PS described in this work reflects what has been described about the distribution of rhythms across the scalp in young people and adults [13,14,17,19] and about PDR frequency and amplitude [18,24,38], but it is more difficult to find equivalence with the results obtained from unipolar databases. Regarding synchronization, only Cohα was age-dependent, although the topographical structure was different for different age groups. Perhaps the limited number of subjects studied is at the root of this lack of dependence on age.

Our main goal was to assess the feasibility of a normative structure of EEG obtained from daily routine practice in a clinical department with nearly 2000 EEGs/year, which means an average of 8 EEGs/day, including external, emergency, hospital-admitted, and ICU patients. Under these conditions, not all recommendations about the density of electrodes, environmental noise, or impedances [47,78,79,80] can be strictly followed because they must achieve a trade-off between good quality and reliability in diagnosis and efficiency for the health system. The introduction of qEEG tools in daily practice, therefore, must be easy and friendly enough to increase the diagnostic power of conventional EEG without increasing the time spent in patient preparation and analyses of recordings [81].

There are some limitations in our study that should be corrected in the future. The first is the limited number of patients. This fact implies that the statistical power of most of our tests was below 0.8. The low value of statistical power (the probability to avoid a type II error) occurs when the sample size is too small and means that we must interpret negative findings cautiously. Therefore, we must be conscious that it is possible that we incorrectly accepted the null hypothesis. Nevertheless, because both type I and II errors depend on a sample size, the risk of a type I error is reduced when the sample size is smaller; therefore, the statistically significant results must be confidently accepted [82]. Obviously, both type of errors, I and II, will be lowered when the number of subjects increases. The use of datasets covering a broader range of ages and populations around the world would increase the accuracy of the normative equations and structure relationships described here. The grouping method in lobes can also be considered a limitation because it is difficult to compare the topographical results obtained in normative unipolar databases. However, this would depend on whether clinicians change from the most extended montage (bipolar) to some that are unipolar (AR, ML, or even REST) in the future.

### 4.3. Future Works

The approach used in this paper is robust, simple, and efficient in discriminating between several pathological situations, e.g., epilepsy, encephalopathy, stroke, coma, and psychiatric illnesses. The topographical method is useful because it yields the lobar/hemispheric identification of the injury, orienting the clinician to the best treatment. Obviously, there are several assumptions in our method that can be debated, such as the use of bipolar montage, the equivalence of results with unipolar ones, or the meaning of mean lobar synchronization obtained from coherence between bipolar channels. More research is needed to answer these and other related questions.

A pilot study can ask whether something can be done should the researchers proceed with it, and if so, how. It would be conducted on a smaller scale than a main or full-scale study [83] and, under these restrictions, we have shown the feasibility of the normative structure of daily rsEEG in clinical practice. The use of a big number of EEG recordings from datasets can be an easy, fast, and straightforward option to answer all the questions opened in this work and a way to achieve a more precise definition of the equations described here.

## 5. Conclusions

The structure of the routine daily practice rsEEG recorded with double-banana bipolar montage can be easily described for PS, synchronization, and PDR. Therefore, this work shows the feasibility of a method currently used in daily routine practice that is friendly for clinicians. The normative structure can serve as a hypersphere defining the range of physiology as a function of age. Values either exceeding or defecting the hypersphere can determine pathological states. In this way, we can expand on overwhelming the spectrum of potential diagnoses by rsEEG, also increasing exactness by means of numerical ranges. Our method is easy to implement in the same computer where EEG is picked up. Therefore, practically every neurophysiologist could benefit from obtaining numerical values during routine clinical practice, which can expand the realm of diagnoses and their severity and yield an objective comparison of serial EEG during large periods of monitoring in ICU. 

## Figures and Tables

**Figure 1 brainsci-13-00167-f001:**
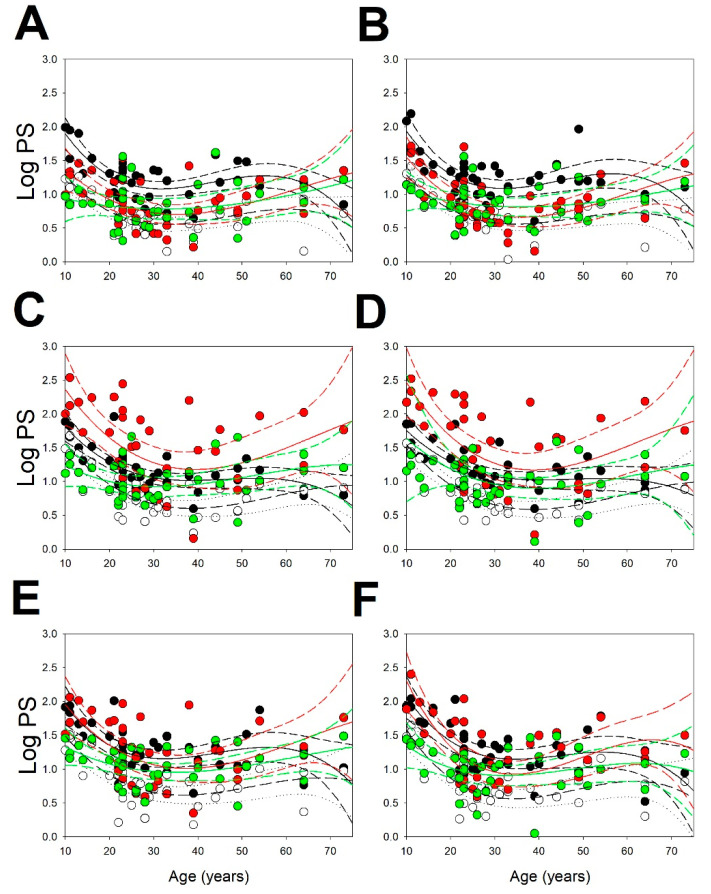
Scatter plots for logPS as function of age. (**A**) Left frontal lobe; (**B**) right frontal lobe; (**C**) left parieto-occipital lobe; (**D**) right parieto-occipital lobe; (**E**) left temporal lobe; and (**F**) right temporal lobe: Black dots = delta band; white dots = theta band; red dots = alpha band; and green dots = beta band. Black and short dashed lines = regression function and 95% interval coefficient for delta; medium dashed and dotted lines = regression function and 95% interval coefficient for theta; red solid and dashed lines = regression function and 95% interval coefficient for alpha; and green solid and dashed lines = regression function and 95% interval coefficient for beta.

**Figure 2 brainsci-13-00167-f002:**
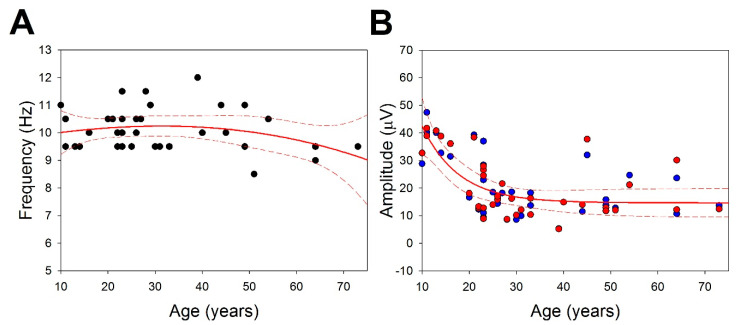
Features of PDR. (**A**) Peak frequency of PDR and regression function (solid red line) with 95% interval confidence (dashed lines); (**B**) Amplitude for the left (blue dots) and right (red dots) parieto-occipital channels. Negative exponential regression functions overlapped: red (r = 0.6890) and blue (r = 0.7011) for right and left hemispheres, respectively.

**Figure 3 brainsci-13-00167-f003:**
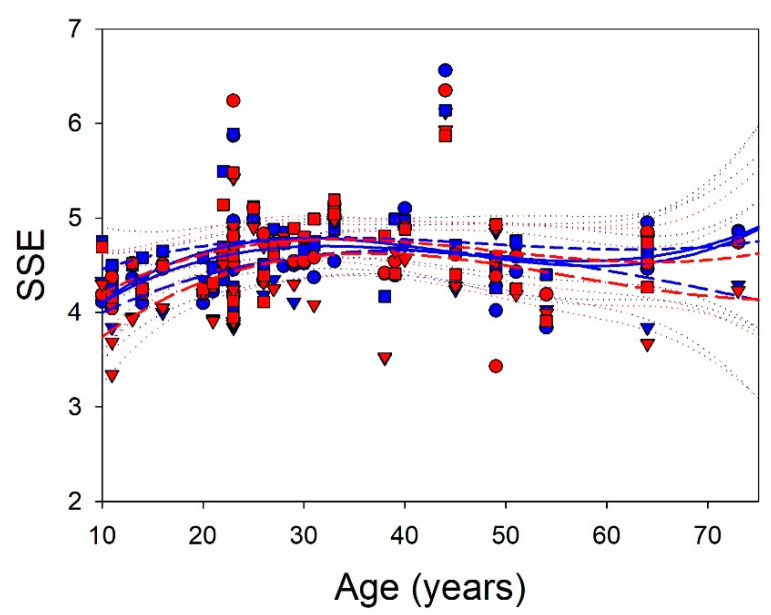
Relationship between SSE and age. Dots = frontal; triangles = parieto-occipital; squares = temporal; dashed lines = quadratic regression functions; dotted lines = 95% interval coefficient; blue = left; red = right.

**Figure 4 brainsci-13-00167-f004:**
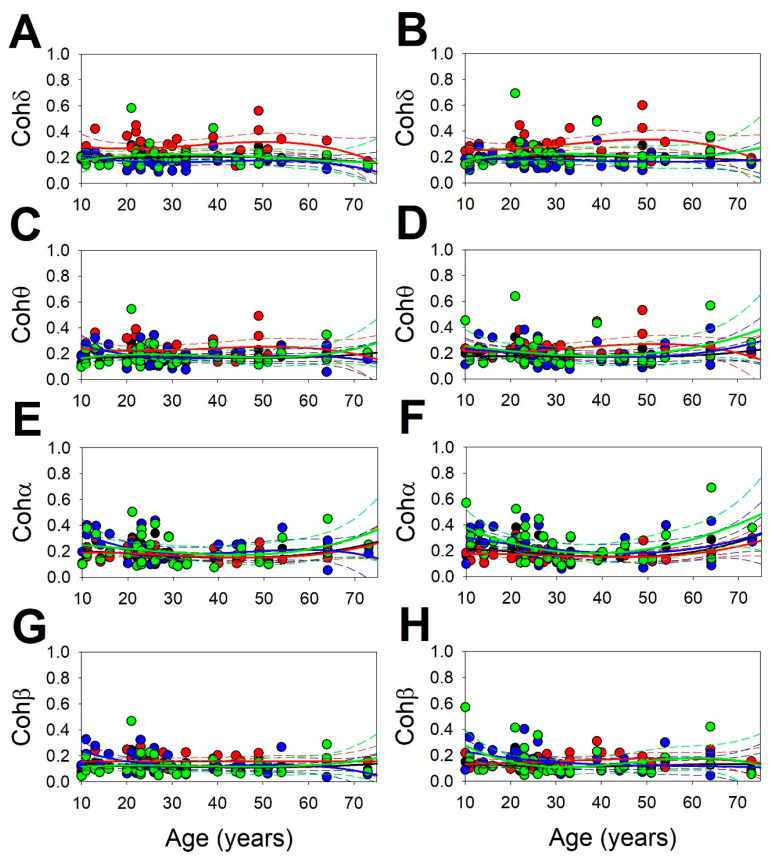
Scatter plot showing the values of synchronization measures as function of age. Regression lines superimposed. (**A**) Left Cohδ; (**B**) Right Cohδ; (**C**) Left Cohθ; (**D**) Right Cohθ; (**E**) Left Cohα; (**F**) Right Cohα; (**G**) Left Cohβ; (**H**) Right Cohβ. Dots and solid lines = H; red dots and red lines = F; blue dots and blue lines = PO; green dots and green lines = T. Continuous lines = quadratic regression functions; dashed lines = 95% coefficient interval.

**Table 1 brainsci-13-00167-t001:** Numerical values for the logPS of bands and lobes. Statistical comparison of symmetric lobes by paired Student’s *t*-test (*N* = 37).

Lobe	Band	Left Hemisphere		Right Hemisphere		*p*
		Mean	SEM	Mean	SEM	
F	Delta	1.24	0.05	1.26	0.06	0.379
	Theta	0.72	0.05	0.72	0.04	0.854
	Alpha	0.90	0.06	0.91	0.06	0.424 *
	Beta	0.85	0.05	0.89	0.05	0.068
PO	Delta	1.18	0.05	1.19	0.05	0.456
	Theta	0.85	0.06	0.85	0.06	0.556
	Alpha	1.52	0.10	1.53	0.10	0.400
	Beta	1.02	0.05	0.99	0.05	0.291 *
T	Delta	1.31	0.06	1.33	0.06	0.490
	Theta	0.84	0.06	0.80	0.06	0.084
	Alpha	1.28	0.08	1.27	0.09	0.526 *
	Beta	1.07	0.05	1.02	0.05	0.161

* Wilcoxon signed-rank test.

**Table 2 brainsci-13-00167-t002:** Structure of the different lobes as function of groups of age. Data are ordered from the highest to lowest value of logPS for every lobe; therefore, the labels do not follow the same order.

Lobe	<20 YEARS (*N* = 7)					20–50 Years (*N* = 25)					>50 Years (*N* = 5)				
		Left		Right			Left		Right			Left		Right	
	Band	Mean	SEM	Mean	SEM	Band	Mean	SEM	Mean	SEM	Band	Mean	SEM	Mean	SEM
F	Delta	1.64	0.11	1.67	0.13	Delta	1.16	0.05	1.18	0.06	Delta	1.11	0.11	1.08	0.05
	Alpha	1.21	0.07	1.30	0.12	Beta	0.81	0.07	0.86	0.06	Alpha	1.09	0.11	1.06	0.14
	Theta	1.10	0.05	1.09	0.09	Alpha	0.77	0.07	0.77	0.06	Beta	0.96	0.11	0.90	0.09
	Beta	0.90	0.03	0.98	0.05	Theta	0.60	0.03	0.63	0.04	Theta	0.78	0.18	0.68	0.12
P-O	Alpha	2.07	0.11	2.15	0.10	Alpha	1.35	0.12	1.35	0.11	Alpha	1.57	0.22	1.58	0.25
	Delta	1.57	0.09	1.56	0.09	Delta	1.10	0.05	1.11	0.05	Beta	1.13	0.09	1.00	0.15
	Theta	1.28	0.12	1.27	0.11	Beta	0.98	0.07	0.95	0.07	Delta	1.00	0.11	1.05	0.09
	Beta	1.12	0.07	1.14	0.07	Theta	0.73	0.06	0.72	0.06	Theta	0.87	0.04	0.88	0.05
T	Alpha	1.79	0.07	1.93	0.14	Delta	1.20	0.05	1.21	0.06	Alpha	1.41	0.14	1.36	0.12
	Delta	1.71	0.07	1.83	0.06	Alpha	1.11	0.09	1.06	0.09	Delta	1.29	0.19	1.22	0.23
	Beta	1.27	0.05	1.24	0.08	Beta	1.00	0.06	0.96	0.07	Beta	1.15	0.11	0.95	0.10
	Theta	1.27	0.10	1.28	0.12	Theta	0.71	0.05	0.66	0.06	Theta	0.87	0.12	0.83	0.15

F = frontal; P-O = parieto-occipital; T = temporal.

**Table 3 brainsci-13-00167-t003:** Structure of the synchronization of different lobes as function of age. Data are ordered from the highest to lowest values of different measurements of synchronization (Coh) lobe; therefore, the column Coh do not follow the same order.

	<20 years						20–50 Years						>50 Years					
	Left			Right			Left			Right			Left			Right		
	Coh	Mean	SEM	Coh	Mean	SEM	Coh	Mean	SEM	Coh	Mean	SEM	Coh	Mean	SEM	Coh	Mean	SEM
H	Cohα	0.1871	0.0079	Cohα	0.2020	0.0058	Cohδ	0.2062	0.0180	Cohδ	0.201	0.0156	Cohα	0.2004	0.0230	Cohα	0.2232	0.0300
	Cohδ	0.1792	0.0068	Cohδ	0.1745	0.0061	Cohθ	0.1668	0.0133	Cohθ	0.169	0.0185	Cohδ	0.1803	0.0067	Cohδ	0.1890	0.0103
	Cohθ	0.1690	0.0124	Cohθ	0.1692	0.0131	Cohα	0.1376	0.0185	Cohα	0.145	0.0108	Cohθ	0.1734	0.0124	Cohθ	0.1854	0.0143
	Cohβ	0.1135	0.0072	Cohβ	0.1261	0.0062	Cohβ	0.1041	0.0050	Cohβ	0.104	0.0059	Cohβ	0.1217	0.0174	Cohβ	0.1351	0.0189
F	Cohδ	0.2711	0.0370	Cohδ	0.2878	0.0185	Cohδ	0.2654	0.0327	Cohδ	0.2486	0.0182	Cohδ	0.294	0.0216	Cohδ	0.2544	0.0348
	Cohθ	0.2276	0.0340	Cohθ	0.2328	0.0173	Cohθ	0.2100	0.0212	Cohθ	0.205	0.0168	Cohθ	0.2366	0.0198	Cohθ	0.1986	0.0246
	Cohα	0.1863	0.0271	Cohα	0.1716	0.0115	Cohα	0.1722	0.0231	Cohα	0.1646	0.016	Cohα	0.1679	0.0118	Cohα	0.1706	0.0344
	Cohβ	0.1630	0.0243	Cohβ	0.1581	0.0108	Cohβ	0.1348	0.0162	Cohβ	0.1283	0.0169	Cohβ	0.1575	0.0117	Cohβ	0.1532	0.0240
PO	Cohα	0.2880	0.0412	Cohα	0.1948	0.0187	Cohα	0.2020	0.0595	Cohα	0.2883	0.0411	Cohα	0.2041	0.0204	Cohα	0.2658	0.0672
	Cohθ	0.2331	0.0301	Cohθ	0.1738	0.0135	Cohθ	0.1760	0.0378	Cohθ	0.2266	0.0309	Cohθ	0.1821	0.0152	Cohθ	0.2376	0.0472
	Cohβ	0.1953	0.0348	Cohδ	0.1697	0.0098	Cohδ	0.1534	0.0167	Cohβ	0.2026	0.0321	Cohδ	0.1731	0.0132	Cohδ	0.1856	0.0201
	Cohδ	0.1910	0.018	Cohβ	0.1314	0.0131	Cohβ	0.1204	0.0446	Cohδ	0.1827	0.0235	Cohβ	0.1442	0.0166	Cohβ	0.1376	0.0507
T	Cohα	0.2140	0.039	Cohδ	0.2238	0.0194	Cohα	0.2596	0.0588	Cohα	0.3120	0.0478	Cohδ	0.2236	0.0244	Cohα	0.3330	0.1010
	Cohδ	0.1686	0.0102	Cohα	0.1940	0.0227	Cohθ	0.2110	0.0367	Cohθ	0.2110	0.0423	Cohα	0.2117	0.0237	Cohθ	0.2644	0.0799
	Cohθ	0.154	0.0218	Cohθ	0.1907	0.0180	Cohδ	0.1870	0.0132	Cohβ	0.1924	0.0646	Cohθ	0.2001	0.0235	Cohδ	0.2314	0.0350
	Cohβ	0.0976	0.0179	Cohβ	0.1265	0.0178	Cohβ	0.1492	0.0402	Cohδ	0.1647	0.0084	Cohβ	0.131	0.0185	Cohβ	0.172	0.0668

H = hemisphere; F = frontal; P-O = parieto-occipital; T = temporal.

## Data Availability

The MATLAB^®^ script is available upon reasonably request from corresponding author.

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
