# Peer review of "Normative Structure of Resting-State EEG in Bipolar Derivations for Daily Clinical Practice: A Pilot Study"

_brainsci, 2023, doi:10.3390/brainsci13020167_

Round 1

Reviewer 1 Report

Dear Authors; I found this pilot study an interesting work on the Normative structure of the resting state EEG in bipolar derivations for daily clinical practice. Prior to processing it further , it needs some extra work. Regards. P.S.

[1] Writing:

1-1 Add list of used abbreviations in the work right before reference section for readers referral.

1-2 Make references in MDPI format. They are written in messy format.

1-3 Make clear the trend of the "4.Discussion" section more clear. Its hard to follow.  Break it down to the following subsections: 4.1. Summary & Contributions; 4.2. Strength & Limitations; 4.3. Future Work.

1-4 Section 2.3 in "3. Materials & Methods" is missing. Add it !

1-5 Missing outline of the study: In line 55-67 the study outline is missing. Add it.

1-6 Numerize your math/stat equations in the text body and in the appendix: line 108, 110, etc. 

[2] Statistical:

2-1 Cite statistical/Mathematical softwares with their versions "clearly" in the reference section. For example,. in line 141 which version was used ? add all these information for the readers  for reproducibility purposes. 

2-2 Line 153 formulae: a subscript  "j" is needed for inside the square root formulae. Add it.

2-3 Low sample size: The study sample size are relatively low: 10, 27. Report statistical power for the tests conducted. Need to be minimum 80%. 

Author Response

Dear reviewer,

Thank you for the comments concerning our manuscript. Those comments are all valuable and very helpful for revising and improving our paper, as well as important guiding significance to our research. We have studied your comments carefully and have made correction which we hope meet your approval. We will reply one by one as follows and the part that replies to you is marked in red of all revisions in the revised manuscript. 

Dear Authors; I found this pilot study an interesting work on the Normative structure of the resting state EEG in bipolar derivations for daily clinical practice. Prior to processing it further , it needs some extra work. Regards. P.S.

Thank you very much for the valuation of our work.

[1] Writing:

1-1 Add list of used abbreviations in the work right before reference section for readers referral.

We have added a list with used abbreviations at the end of the work.

1-2 Make references in MDPI format. They are written in messy format.

Done

1-3 Make clear the trend of the "4.Discussion" section more clear. Its hard to follow.  Break it down to the following subsections: 4.1. Summary & Contributions; 4.2. Strength & Limitations; 4.3. Future Work.

Thanks for this observation. We have modified Discussion following your suggestion.

1-4 Section 2.3 in "3. Materials & Methods" is missing. Add it !

Done.

1-5 Missing outline of the study: In line 55-67 the study outline is missing. Add it.

Thank you again for this comment. We have modified the Introduction accordingly explaining this point.

1-6 Numerize your math/stat equations in the text body and in the appendix: line 108, 110, etc. 

Done

[2] Statistical:

2-1 Cite statistical/Mathematical softwares with their versions "clearly" in the reference section. For example,. in line 141 which version was used ? add all these information for the readers  for reproducibility purposes. 

We have added R2018 (Natik, Massachusetts, USA).

2-2 Line 153 formulae: a subscript  "j" is needed for inside the square root formulae. Add it.

Sorry but we have checked the formula (6), line 163 and we think that it is correct, because the j refers to the subscript of Vj (left or rigth), meanwhile the summatory inside the square root is onto the index i.

2-3 Low sample size: The study sample size are relatively low: 10, 27. Report statistical power for the tests conducted. Need to be minimum 80%.

Thank you again for this important question. We have added this paragraph at Discussion “This fact implies that the statistical power of most of our test were below 0.8. The low value of statistical power (the probability to avoid a Type II error) occurs when the sample size is too small and means that we must interpret the negative findings cautiously. Therefore, we must be conscious that it is possible that we incorrectly accept the null hypothesis. Nevertheless, because both type I and II errors depend on the sample size, the risk of a type I error is reduced when the sample size becomes smaller and, therefore the statistically significant results must be confidently accepted [85]. Obviously, both type of errors, I and II will be lowered when the number of subjects increase”

Reviewer 2 Report

In the present research article entitled ‘Normative structure of the resting state EEG in bipolar derivations for daily clinical practice. A pilot study’ Pastor and Vega-Zelaya analyzed the feasibility to build a normative structure of the resting state EEG in healthy people using the common conditions applied in daily clinical practice. For this purpose, authors have obtained the structure of bands for a double-banana bipolar montage, the PDR features and the synchronization by lobes and hemispheres using coherence. The results depicted the physiological structure for EEG form healthy humans for the most used montage in daily clinical practice.

The main strength of this manuscript is that it addresses an interesting and timely question, and as a pilot study it describes the feasibility for a normative structure of rsEEG picked up with a double banana montage. In general, I think the idea of this article is really interesting and the authors’ fascinating observations on this timely topic may be of interest to the readers of Brain Sciences. However, some comments, as well as some crucial evidence that should be included to support the author’s argumentation, needed to be addressed to improve the quality of the manuscript, its adequacy, and its readability prior to the publication in the present form, in particular reshaping parts of the Introduction and Methods sections by adding more evidence and theoretical constructs.

Please consider the following comments:

A graphical abstract that will visually summarize the main findings of the manuscript is highly recommended.

Keywords: I suggest changing the keyword ‘double banana’ in ‘double banana bipolar montage’, which in my opinion seems to be more appropriate.

Abstract: According to the Journal’s guidelines, the abstract should be a total of about 200 words maximum. Please correct the actual one.

Introduction: The ‘Introduction’ section well-written and nicely presented, with a detailed description of different available EEG montages used during a recording session for clinical interpretation. Nevertheless, I believe that here authors could focus on description of common double-banana montage in Psychological research: for example adding evidence of quantitative assessment with longitudinal montage, as some specific measure, like frontal lobe alpha activity, is highlighted in the quest for physiological correlates of attention and perception (https://doi.org/10.3390/biomedicines10123189), also in psychiatric populations ( https://doi.org/10.1038/s41598-022-06503-1). 

Subjects: Data about participants are not adequately explained. Could the authors specify how did they estimate the exact number of participants? Did they use a power analysis?

Results: I suggest rewriting this section more accurately. To properly present experimental findings, I think that authors should provide full statistical details (like degree of freedom or post-hoc utilized), to ensure in-depth understanding and replicability of the findings.

In my opinion, I think the ‘Conclusions’ paragraph would benefit from some thoughtful as well as in-depth considerations by the authors, because as it stands, it is very descriptive but not enough theoretical as a discussion should be. Authors should make an effort, trying to explain the theoretical implication as well as the translational application of their research.

In according to the previous comment, I would ask the authors to include a proper and defined ‘Limitations and future directions’ section before the end of the manuscript, in which authors can describe in detail and report all the technical issues brought to the surface.

References: Authors should consider revising the bibliography, as there are several incorrect citations. Indeed, according to the Journal’s guidelines, they should provide the abbreviated journal name in italics, the year of publication in bold, the volume number in italics for all the references. Also, please correct in-text citations: reference should be numbered, and placed in square brackets [ ] (for example [1]).

Overall, the manuscript contains 4 figures, 3 tables and 85 references. I believe that the manuscript might carry important value in describing the feasibility for a normative structure of rsEEG picked up with a double banana montage.

I am available for a new round of revision of this paper. I declare no conflict of interest regarding this manuscript. 

Best regards,

Reviewer

Author Response

Dear reviewer:

Thank you for the comments concerning our manuscript. Those comments are all valuable and very helpful for revising and improving our paper, as well as the important guiding significance to our researches. We have studied comments carefully and have made correction which we hope meet with approval. We will reply one by one as follows and the part that replies to you is marked in blue of all revisions in the revised manuscript.

In the present research article entitled ‘Normative structure of the resting state EEG in bipolar derivations for daily clinical practice. A pilot study’ Pastor and Vega-Zelaya analyzed the feasibility to build a normative structure of the resting state EEG in healthy people using the common conditions applied in daily clinical practice. For this purpose, authors have obtained the structure of bands for a double-banana bipolar montage, the PDR features and the synchronization by lobes and hemispheres using coherence. The results depicted the physiological structure for EEG form healthy humans for the most used montage in daily clinical practice.

The main strength of this manuscript is that it addresses an interesting and timely question, and as a pilot study it describes the feasibility for a normative structure of rsEEG picked up with a double banana montage. In general, I think the idea of this article is really interesting and the authors’ fascinating observations on this timely topic may be of interest to the readers of Brain Sciences. However, some comments, as well as some crucial evidence that should be included to support the author’s argumentation, needed to be addressed to improve the quality of the manuscript, its adequacy, and its readability prior to the publication in the present form, in particular reshaping parts of the Introduction and Methods sections by adding more evidence and theoretical constructs.

Thank you very much for your opinion about our work

Please consider the following comments:

  • A graphical abstract that will visually summarize the main findings of the manuscript is highly recommended.

Done.

  • Keywords: I suggest changing the keyword ‘double banana’ in ‘double banana bipolar montage’, which in my opinion seems to be more appropriate.

Done.

  • Abstract: According to the Journal’s guidelines, the abstract should be a total of about 200 words maximum. Please correct the actual one.

Thank you very much for this comment. We have corrected the abstract to 199 words.

  • Introduction: The ‘Introduction’ section well-written and nicely presented, with a detailed description of different available EEG montages used during a recording session for clinical interpretation. Nevertheless, I believe that here authors could focus on description of common double-banana montage in Psychological research: for example adding evidence of quantitative assessment with longitudinal montage, as some specific measure, like frontal lobe alpha activity, is highlighted in the quest for physiological correlates of attention and perception (https://doi.org/10.3390/biomedicines10123189), also in psychiatric populations (https://doi.org/10.1038/s41598-022-06503-1). 

Thank you again for drawing our attention to these very recents articles. We have included it at Discussion.

  • Subjects: Data about participants are not adequately explained. Could the authors specify how did they estimate the exact number of participants? Did they use a power analysis?

You are absolutly right. As stated as Introductrion, our main goal was to assess the feasability of normative equations in a broad range of age. We are conscious that power is very low (clearly lower than 0.8), but we have shown that it is posible to describe specific topographycal equations for different bands along the age. We asume that these preliminary results would be not the definitive, because we need to increase the number of subjects, probably obtaining recordings from datasets.

  • Results: I suggest rewriting this section more accurately. To properly present experimental findings, I think that authors should provide full statistical details (like degree of freedom or post-hoc utilized), to ensure in-depth understanding and replicability of the findings.

Dear reviewer, we are not sure what do you mean, because the degrees of freedom are indicated at lines 141, 358. Posthoc test were not used because we did not use ANOVA.

  • In my opinion, I think the ‘Conclusions’ paragraph would benefit from some thoughtful as well as in-depth considerations by the authors, because as it stands, it is very descriptive but not enough theoretical as a discussion should be. Authors should make an effort, trying to explain the theoretical implication as well as the translational application of their research.

Thank you for this very clever observation. We have added this paragraph at Conclusión: “ The normative structure can serve as a hypersphere defining the range of physiology as a function of age. Values either exceeding or defecting hypersphere can determine pathological states. In this way, we can expand overwhelming the spectrum of potential diagnosis by rsEEG, increasing also the exactness by means of numerical ranges. Our method is easy to implement in the same computer where EEG is picked-up. Therefore, practically every neurophysiologist could benefit from obtaining numerical values during its routine clinical practice, which can expand several orders de magnitude the realm of diagnosis, its severity and allowing an objective comparison during serial EEG or during large periods of monitoring in ICU”

  • In according to the previous comment, I would ask the authors to include a proper and defined ‘Limitations and future directions’ section before the end of the manuscript, in which authors can describe in detail and report all the technical issues brought to the surface.

Done

  • References: Authors should consider revising the bibliography, as there are several incorrect citations. Indeed, according to the Journal’s guidelines, they should provide the abbreviated journal name in italics, the year of publication in bold, the volume number in italics for all the references. Also, please correct in-text citations: reference should be numbered, and placed in square brackets [ ] (for example [1]).

 Corrected

Overall, the manuscript contains 4 figures, 3 tables and 85 references. I believe that the manuscript might carry important value in describing the feasibility for a normative structure of rsEEG picked up with a double banana montage.

I am available for a new round of revision of this paper. I declare no conflict of interest regarding this manuscript. 

Best regards,

Round 2

Reviewer 2 Report

The authors did an excellent job clarifying all the questions I have raised in my previous round of review. Currently, this paper is a well-written, timely piece of research that explored the feasibility to build a normative structure of the resting state EEG in healthy people using the common conditions applied in daily clinical practice.

Overall, this is a timely and needed work. It is well-researched and nicely written, therefore I believe that this paper does not need a further revision and meets the Journal’s high standards for publication.

I am always available for other reviews of such interesting and important articles. Thank You for your work,

Reviewer